# XiEff Representation for Near-Field Optics

## Abstract

Near-field optics, or near-field electrodynamics, is a field that studies the interaction between materials and light at spatial scales smaller than the wavelength. At these extremely small scales, below the diffraction limit, the interaction between materials and electromagnetic fields can exhibit unique behaviors and properties not observed in conventional optics. This area of research is crucial for understanding the optical characteristics of nanotechnical systems and nanoscale biological objects. One of the primary tools used in near-field optics research is scanning near-field optical microscopy (SNOM), which allows researchers to measure near-field optical images (NFI). However, these images often lack visual clarity and interpretability, hindering a comprehensive understanding of the properties of the probed particles.

The main goal of this paper is to introduce a novel approach that addresses these challenges. Inspired by the progress in physics-informed neural networks (PINNs) and its applied subdomain physics-informed computer vision (PICV), we propose an unsupervised method that introduces the XiEff representation – a neural field-based parameterization of the effective susceptibility tensor. By integrating XiEff into the Lippmann-Schwinger integral equation framework for near-field optics, we develop an optimization strategy to reconstruct the effective susceptibility distribution directly from NFI data.

The optimized XiEff representation provides an interpretable and explainable model of the particle's shape. Extensive evaluations on a synthetically generated NFI dataset demonstrate the effectiveness of the method, achieving high intersection-over-union scores between XiEff and ground truth shapes, even for complex geometries. Furthermore, the approach exhibits desirable robustness to measurement noise, a crucial property for practical applications. The XiEff representation, combined with the proposed optimization framework, potentially introduces a valuable tool for enabling explainable near-field optics imaging and enhancing the understanding of particle characteristics through interpretable representations

## 1 Introduction

Near-field optics Girard & Dereux (1996) is a branch of optical physics that explores the behavior of light at scales smaller than the diffraction limit of conventional optics, where unique optical phenomena arise due to near-field interactions. Scanning near-field optical microscopy Durig et al. (1986) (SNOM) is a powerful technique that employs a nanoscopic probe to measure near-field optical signals with exceptional spatial resolution, providing unprecedented access to optical properties at the nanoscale.

Despite SNOM's advanced capabilities, the raw near-field images (NFIs) it produces often lack visual clarity and interpretability, making it challenging to infer the physical properties and geometry of the probed particles. This limitation hinders a comprehensive understanding of near-field optical interactions and restricts the potential applications of SNOM.

To address this challenge, we propose a novel approach that introduces the XiEff representation — a neural field-based parameterization Xie et al. (2022) of the effective susceptibility tensor $\hat{X}$ Lozovski (2010). Our method leverages recent advancements in Physics-Informed Neural Networks (PINNs) Raissi et al. (2019) and their applied subdomain, Physics-Informed Computer Vision

(PICV) Banerjee et al. (2024). PINNs integrate physical laws into neural network training by embedding partial differential equations (PDEs) and boundary conditions into the loss functions, guiding the learning process with physics-based constraints. PICV extends these principles to computer vision tasks.

Neural Radiance Fields (NeRFs) Mildenhall et al. (2021), which are a part of PICV, have revolutionized 3D reconstruction and novel view synthesis by representing scenes as neural networks. In our work, we adopt similar concepts but focus on representing the effective susceptibility tensor as neural fields tailored for near-field optics.

We integrate the XiEff representation into the Lippmann-Schwinger integral equation Girard et al. (1995) framework, which utilizes Green's functions to describe electromagnetic field interactions. This approach allows us to incorporate physical laws directly into our neural network without relying explicitly on PDEs or boundary conditions, making it suitable for practical problems where such information may be inaccurate or unknown.

The main contributions of this work are:

1. **Novel Computer Vision Algorithm** We introduce the XiEff representation, a new computer vision algorithm that uses neural fields to model the effective susceptibility tensor in near-field optics imaging. Our method reconstructs the effective susceptibility XiEff directly from near-field imaging data. This representation provides an interpretable model connected to the optical characteristics of the object. It offers meaningful physical insights that align with human perception, making the object's optical characteristics understandable, unlike the raw near-field images that are hard to interpret.

2. **Successful Validation** Our method was validated through extensive synthetic data experiments. The XiEff representation achieved high IoU scores with particles shapes, showed robustness to noise, and was computationally efficient and hyperparametrs free. It successfully reconstructs simple and complex particle shapes, maintaining reliable convergence without requiring labeled data or external data.

3. **SNOM Scanning Strategy** We propose a SNOM scanning method using randomly varied external fields to enhance data diversity, leading to more accurate reconstructions. This strategy is potentially universal, capable of improving various inverse algorithms in similar domains.

4. **Benchmark Dataset** We created a comprehensive dataset of particle shapes and their NFIs, to be released as the first benchmark for inverse problems in near-field optics and PINNs

In summary, we aim to expand computer vision into the new modality of near-field optics by enhancing interpretability and effectiveness of near-field imaging.

## 2 RELATED WORK

Physics-informed neural networks (PINNs) have emerged as a powerful framework for solving complex physical systems by incorporating physical laws, typically in the form of partial differential equations (PDEs), into neural network loss functions Raissi et al. (2019). While PINNs have shown remarkable success across various domains, comprehensive surveys Huang et al. (2022); Banerjee et al. (2024); Wang et al. (2024b;a) indicate that their applications have primarily focused on PDEs, despite many physical systems having correspondents integral formulations. The integration of integral equations within the PINNs framework remains relatively unexplored, with only a few theoretical studies addressing this approach Sun et al. (2023).

An applied extension of PINNs, Physics-Informed Computer Vision (PICV) Banerjee et al. (2024), has successfully incorporated physical laws into computer vision algorithms. Our work on near-field imaging extends the PICV paradigm by integrating electromagnetic theory into computer vision algorithm to enable interpretable reconstruction of particle characteristics from SNOM measurements. Like other PICV applications, our work confronts several key challenges common in PICV Banerjee et al. (2024), including balancing physics and data constraints, selecting appropriate physics priors, developing standardized datasets and benchmarks, computational efficency, and maintaining interpretability and explainability.

In the context of near-field problems, the conventional approach involves Maxwell's equations, which comprise four fundamental equations and up to six boundary conditions Jackson (2012). When implementing these within PINNs, significant challenges arise in loss function balancing Wang et al. (2021) and handling boundary conditions, particularly in inverse problems where the target shape needs to be reconstructed. The Lippmann-Schwinger (LS) equation offers an attractive alternative for such inverse problems, as it inherently handles boundary conditions through its integral formulation.

Recent works in near-field optics has demonstrated the application of PINNs based on full-vector Maxwell's PDEs for solving inverse problems Chen et al. (2020); Chen & Dal Negro (2022). The fist approach Chen et al. (2020) enabled the inverse retrieval of scalar photonic nanostructure properties, such as electric permittivity and magnetic permeability, from near-field data. Subsequently, the second approach Chen & Dal Negro (2022) reconstructed spatial material characteristics of a spherical object. However, this experiment relied on measurements of the field distribution directly near the object's surface (which shape's mirrored the particle's shape), and required substantial computational resources (10 hours on a 1080Ti GPU, 50×50×30 computational grid). Both research efforts were constrained to investigating objects with very simple geometries in 2D case, and for 3D case inverse problem reported only single experiment with spherical object.

A notable example of applying PINNs in the domain of diffraction tomography is the work Saba et al. (2022). In this work, a UNet-based physics-informed neural network is used as a forward solver, trained on a synthetic dataset of biological cell objects. The forward solver is then employed in an inverse reconstruction of the refractive index (RI) by minimizing the discrepancy between predicted and measured scattered fields. Despite being trained mainly on synthetic data, this method demonstrates good performance on real datasets. However, a limitation of this approach is that it performs best for object geometries similar to those present in the synthetic dataset. The proposed method also shows good computational efficiency (4.5 min, 64x64x64 computational grid).

In the radio frequency domain Zhang et al. (2024) utilized the integral Lippmann-Schwinger equation with PINNs for modeling electromagnetic scattering in complex 2D shapes. However, their focus remained on forward problem solving rather than inverse problems.

Neural Radiance Fields (NeRF) Xie et al. (2022) have achieved remarkable success in computer vision and scene rendering. Interestingly, NeRF can be interpreted as an integral equation where boundary conditions are explicitly encoded, effectively serving as a phenomenological physical model. However, NeRF's novel view interpolations often exhibit visual inconsistencies and geometric roughness. Recent research aims to address this "generalization gap" by incorporating physical principles rather than relying solely on simple phenomenological model Li et al. (2023).

The limited exploration of integral equations in physics-informed machine learning presents an opportunity for innovation. Integral formulations often offer advantages in handling boundary conditions and provide more natural representations for certain physical phenomena then decomposition physical lows into differential equations, boundary conditions and initial conditions. This untapped potential suggests promising directions for future research, particularly in problems both theoretical and applied PINNs.

## 3 THE LIPPMANN-SCHWINGER EQUATION

The problem under consideration can be described as follows. We consider an external electric field and a homogeneous dielectric that interacts with the field. The dielectric undergoes polarization thus creating an induced electric field. Any measuring device outside the dielectric will measure superposition $\vec{E}$ of the initial external field $\vec{E}^{(0)}$ and polarization field $\vec{E}^{\text{pol}}$ (see figure 1). We will try to recover the form of the dielectric basing on the measurements performed at some distance from it.

As a first step we need to express the problem in an analytical way. It is well-known that such matter-field interactions are governed by the Maxwell's equations. But the latter ones are unwieldy, thus we will use the fact that for a homogeneous, linear, non-dispersive, non-magnetic dielectric in an external field, the four Maxwell's equations can be transformed into the Lippman-Schwinger

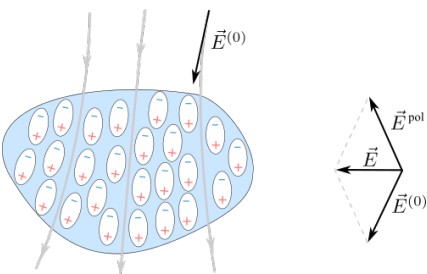

Figure 1: Measurement is performed outside the particle

equation (LS) Girard et al. (1995)

$$\vec{E}(\vec{r},\omega) = \vec{E}^{(0)}(\vec{r},\omega)-$$
$$-\iiint_{\mathbb{R}^3} \hat{\mathcal{G}}(\vec{r},\vec{r}^{'},\omega)\hat{\chi}(\vec{r}^{'},\omega)\vec{E}(\vec{r}^{'},\omega)d\vec{r}^{'} \tag{1}$$

where $\hat{\chi}$ is the local electric susceptibility and $\hat{\mathcal{G}}$ is a dyadic propagator described by a complicated formula

$$\hat{\mathcal{G}} = \left(\frac{4\pi\omega^2}{c^2}\mathbb{I} + \vec{\nabla}\otimes\vec{\nabla}\right)\frac{e^{i\omega|\vec{r}-\vec{r}^{'}|/c}}{|\vec{r}-\vec{r}^{'}|}. \tag{2}$$

It is supposed here that the field may be varying with angular frequency $\omega$ and $c$ is the speed of light. Please note that for vacuum $\hat{\chi} = 0$, thus integration is effectively performed over the volume of the particle only.

For our purposes we suppose that the external field is slowly varying $\omega \to 0$ (so-called near-field approximation that applies to a scanning near-field optical microscopy (SNOM) probe and many other cases), thus (1) and (2) can be simplified and rewritten in Cartesian coordinates in a component-wise form

$$E_i(\vec{r}) = E_i^{(0)}(\vec{r})-$$
$$-\iiint_{\mathbb{R}^3} \mathcal{G}_{ij}(\vec{r},\vec{r}^{'})\chi_{jk}(\vec{r}^{'})E_k(\vec{r}^{'})d\vec{r}^{'} \tag{3a}$$

$$\mathcal{G}_{ij} = \frac{\partial^2}{\partial x_i \partial x_j}\frac{1}{|\vec{r}-\vec{r}^{'}|}. \tag{3b}$$

The undisturbed external field $\vec{E}^{(0)}$ is supposed to be known everywhere, while the actual electric field $\vec{E}$ is known only starting at some distance outside the particle (i.e. we cannot measure inside the particle, neither do we know the form of the particle to measure the field at the boundary).

## 4  EFFECTIVE SUSCEPTIBILITY

Now the problem boils down to the continuation of $\vec{E}$ from the areas it was measured to the whole space. This is a non-trivial task not only due to the complexity of (3), but the fact that $\hat{\chi}$ is unknown as well: if we knew it, we would easily recover the shape of the particle due to the fact that $\hat{\chi} = 0$ for vacuum but not for the particle. To tackle the problem we are using the effective susceptibility Lozovski (2010)

$$X_{jk} = \chi_{jk}\frac{E_k^{(0)}}{E_k}.$$

Please note that in this case no summation over repeating indices is performed.

Now the final form of the LS equation can be written as

$$E_i(\vec{r}) = E_i^{(0)}(\vec{r})-$$
$$-\iiint_{\mathbb{R}^3} \mathcal{G}_{ij}(\vec{r},\vec{r}^{'})X_{jk}(\vec{r}^{'})E_k^{(0)}(\vec{r}^{'})d\vec{r}^{'}. \tag{4}$$

Here the only unknown quantity is the $X$, as the $\vec{E}$ in the left-hand side of the formula is measured outside the particle, for example via SNOM probe, thus known.

In computational physics, several methodologies exist for solving the LS equation (3). The Discretization Paulus & Martin (2001) and Iteration Methods Martin et al. (1994) are traditional methods that are computationally hard and do not handle resonance effectively. The Diagram Method is a semi-analytical approach that uses effective susceptibility abstraction, simplifying computation to a quadrature process. It performs well under resonant conditions and is best suited for shapes with known analytical solutions Bozhevolnyi et al. (2001).

In the next section, we reuse the effective susceptibility concept with PINN paradigm and low computational cost for inference (4) will make inverse modeling efficient.

### 4.1 METHOD

In our method, we employ a neural field to mimic the effective susceptibility tensor $\hat{X}(\mathbf{r})$. This approach is akin to the principles of Physics-Informed Neural Networks (PINNs), where physical laws guide the training of neural networks. Instead of incorporating partial differential equations and boundary conditions explicitly and use excess hyperparameters Chen et al. (2020); Chen & Dal Negro (2022), we integrate the neural representation of the effective susceptibility tensor into the Lippmann-Schwinger equation. As result proposed algorithms is hyperparamers free and is more efficient in tuning. This allows the network to learn a physically consistent representation of the susceptibility tensor directly from NFI data.

The tensor $\hat{X}(\vec{r})$, which is a $3 \times 3$ tensor in three-dimensional space (or $2 \times 2$ in two-dimensional cases), is parameterized as a neural network function $\hat{X}(\vec{r}) = \text{XiEff}(\vec{r}, \Theta)$ with parameters $\Theta$. With this representation we can reformulate the Lippmann-Schwinger equation (4) by simply setting $X_{ij}(\vec{r}) \rightarrow \text{XiEff}_{ij}(\vec{r}, \Theta)$. Now field $\vec{E}$ can be treated as depending from $\Theta$ through XiEff via the equation (4), thus we write $\vec{E}(\vec{r}; \text{XiEff}(\vec{r}; \Theta))$, or $\vec{E}(\vec{r}; \Theta)$ for short.

Measuring the electric field by physical device, say SNOM probe Durig et al. (1986), we can say that at the points $\vec{r}_i$ its value should constitute $\vec{E}_i^{(obs)}$, thus the loss function is expected to be

$$\mathcal{L}(\Theta) = \sum_{i=1}^{N} ||\vec{E}(\vec{r}_i; \Theta) - \vec{E}_i^{(obs)}||^2 + \lambda \mathcal{L}_{\text{reg}}(\Theta).$$ (5)

We expect that after optimization the neural network function $\text{XiEff}(\vec{r}, \Theta)$ which represents the effective susceptibility will approach zero in regions outside the object to align with real-world data: the susceptibility of vacuum is zero. Thus we impose regularization in form of

$$\mathcal{L}_{\text{reg}}(\Theta) = \sum_{i=1}^{N} ||\text{XiEff}(\vec{r}_i, \Theta)||,$$ (6)

where $\lambda$ signifies the regularization strength. This part of loss is optional.

The minimization of the loss function $\mathcal{L}(\Theta)$ is achieved through gradient-based optimization algorithms, such as stochastic gradient descent (SGD). The ultimate expectation is that the morphology of the optimized $\text{XiEff}_{ij}(\vec{r}; \Theta)$ will resemble the form of the physical particle, thereby providing a mechanism for surface reconstruction within the domain of near-field optical imaging and similar setups.

Before we move to the description of experiments, one more concern should be addressed. We want to be sure that the results will be reliable, i.e. they do not change significantly due to a small variation in the input data. This is crucial when working with physical measurements, as every real-world device provides measurement with certain level of noise. Thus we prove:

**Lemma.** If the observed electric field $\vec{E}_i^{(obs)}$ is perturbed by noise $\vec{\xi}$ with a mean of zero $\mathbb{E}[\vec{\xi}] = 0$, the gradient $\vec{\nabla}_{\Theta} \mathcal{L}(\Theta)$ of the loss function (5) with regularization (6) remains unbiased.

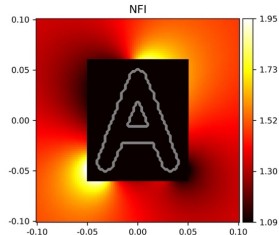 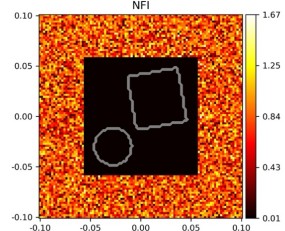

Figure 2: Example of the dataset images without (left) and with (right) random component to the pumping SNOM field. Hidden shape is overlaid for demonstration purposes only and not accessible by NN.

◀ Consider the derivative of the $\vec{\nabla}_\Theta \mathcal{L}(\Theta)$ from (5). The norm squared has form $\vec{x}^T \vec{x}$, thus we can use equality from Petersen et al. (2008) to get

$$\vec{\nabla}_\Theta \mathcal{L}(\Theta) = 2 \sum_{i=1}^{N} \left( \vec{E}(\vec{r}_i; \Theta) - \vec{E}_i^{(obs)} \right) \cdot \vec{\nabla}_\Theta \vec{E}(\vec{r}_i; \Theta) +$$
$$+ \lambda \vec{\nabla}_\Theta \mathcal{L}_{\text{reg}}(\Theta).$$

We can get $\vec{\nabla}_\Theta \mathcal{L}^{(\text{noisy})}(\Theta)$ by simply changing $\vec{E}_i^{(obs)} \to \vec{E}_i^{(obs)} + \vec{\xi}_i$, thus mean of the difference

$$\vec{\nabla}_\Theta \mathcal{L} - \mathbb{E}[\vec{\nabla}_\Theta \mathcal{L}^{(\text{noisy})}] = 2 \sum_{i=1}^{N} \mathbb{E}[\vec{\xi}_i] \cdot \vec{\nabla}_\Theta \vec{E}(\vec{r}_i; \Theta),$$

where we used the linearity of mean to simplify the expression. Now recall that $\mathbb{E}[\vec{\xi}] = 0$ by assumption, thus mean of the difference is zero that concludes the proof. ∎

This simplification shows that the gradient of the loss function with respect to the network parameters $\Theta$ remains unbiased in the presence of noise. Consequently, the optimization via gradient descent is robust, converging to a solution that is not skewed by the noise in the observed data. This robustness is essential for ensuring the reliability of the optimization process in practical applications where measurement noise is inevitable.

## 5  DATASET

In this study, we focus on 2D geometric cases because generating real-world NFI data requires creating nanosized particles with diverse shapes and measuring them, which is a lengthy and costly process. Instead, we chose to create a synthetic 2D dataset, as it is easier to visualize, provides diverse shapes, and typically, NFI images measured by SNOM are also usually 2D, making the research process more efficient. This can be understood as either considering a 2D slice of the system or as a projection of a system that extends infinitely along the Z axis.

We have developed a synthetic dataset, herein referred to as the dataset, consisting of various particles alongside their corresponding calculated near-field images (NFIs). A typical visualization of our dataset you can find at Figure 2. The particle's shape is obscured by an object mask — essentially a rectangular mask that covers the particle with padding — depriving the neural network of any information about the particle's form (previous work Chen & Dal Negro (2022) relied on measurements taken directly near the surface, which resulted in unintended leakage of the object's form due to the measurement method). The corresponding NFI is calculated according to LS solution outside the object mask thus can be measured in a real-life experiment with SNOM probe or other method. Both the $X$ and $Y$ coordinates are normalized to the wavelength, ensuring all NFIs within the range of $-0.1$ to $0.1$. The observed field (NFI), $\vec{E}^{(obs)}$ is also presented in normalized units.

For the initial dataset (see Fig. 2) of near-field images, the external field (e.g. pumping field of SNOM Durig et al. (1986) ) is kept constant at $\vec{E}^{(0)} = (1; 1; 0)$. The dataset encompasses a total of 121 images, including a set of randomly generated simple shapes (17 ellipses, 22 rectangles, 12

Table 1: IoU metrics for different shape categories and dataset variants.

| Dataset | Ellipse | Rectangle | Convex | Union | Character | Avg |
|---------|---------|-----------|--------|-------|-----------|-----|
| constant | 93.51 | 84.08 | 77.02 | 78.65 | 61.70 | 76.72 |
| random | 93.75 | 89.85 | 84.89 | 81.01 | 66.59 | 80.79 |

convex polygons) and more complex forms (36 characters spanning 0-9, A-Z, and 31 union pairs of simple shapes). The susceptibility of the particles was randomly selected from a range of 0.5 to 1.5 and set diagonal and equal for all components. This choice of isotropic susceptibility is well-justified as most common dielectric materials exhibit only weak anisotropic responses at optical frequencies Landau et al. (2013); Boyd et al. (2008). For our 2D geometry, where system symmetry further supports such treatment, this approximation maintains physical relevance while keeping the inverse problem tractable. All scripts involved in generating the dataset and implementing the proposed solution will be made accessible on GitHub.

Additionally, we have developed a second variation of the dataset that utilizes the same shapes and susceptibility but incorporates an external field $\vec{E}^{(0)}$ (pumping field of SNOM is controlled parameter of experiment) that, unlike the previous case, is randomized for each individual SNOM probe position. This approach simulates an ensemble of measurements under varying field conditions, with $\vec{E}^{(0)}$ sampled from a uniform distribution on the square $[-1; 1] \times [-1; 1] \times 0$ for every data point measurement. While this dataset of NFIs may lack some visual clarity, it offers several benefits for the XiEff representation optimization. It enhances optimization robustness by exposing the model to diverse field conditions, helps prevent overfitting to specific configurations, and improves generalizability. This also allows for a more comprehensive performance assessment across various experimental scenarios.

We are also going to release this dataset to the public, and we hope it will serve as a valuable benchmark for assessing approaches in both inverse problems in Physics-Informed Neural Networks (PINNs) and near-field optics.

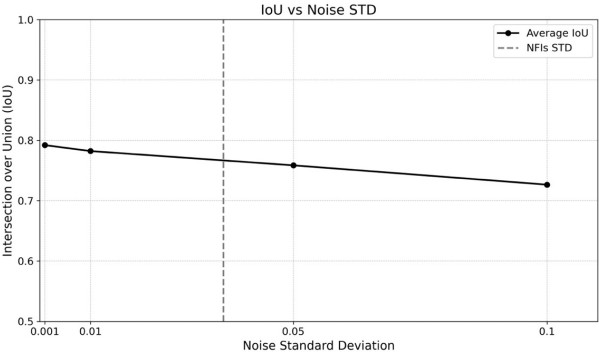

Figure 3: Dependence of IoU on noise level.

## 6 EXPERIMENTS

In this section, we present the details of our experiments and discuss the results. The optimization process was based on the Lippmann-Schwinger (LS) formalism (see (3), (4)), aiming to optimize the XiEff$(\vec{r}; \Theta)$ (see (5), (6)) representation from the near-field imaging (NFI) data. The architecture of XiEff was a multilayer perceptron (MLP), similar to the well-adopted architecture in Neural Radiance Field (NeRF) framework Mildenhall et al. (2021); Xie et al. (2022). The XiEff MLP architecture was constant across all experiments, consisting of 9 layers, each with 256 neurons. The MLP accepted a 2-dimensional input corresponding to the spatial coordinates $(x, y)$, and produced a 4-dimensional output representing the components of the susceptibility tensor $\hat{X}$. Skip connections were added at the 5-th layer, and positional encoding with a frequency of 7 and a *sin* activation. We

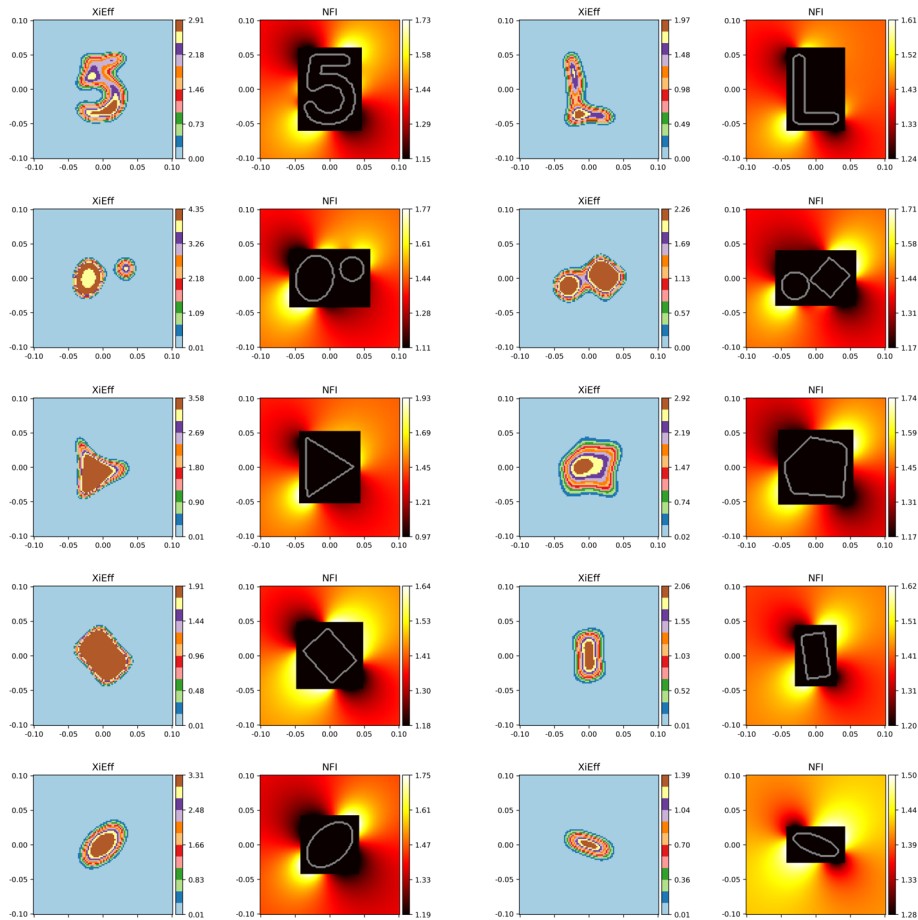

Figure 4: Constant dataset, Xieff and NFI.

employed the Huber loss Huber (1992) as it proved to be more robust than L2 loss. The Huber loss is known to be less sensitive to outliers than the L2 loss, which can be beneficial when dealing with noisy data or observations that may contain some extreme values. Please note that for $\Theta$ close to $\Theta_{\text{optimal}}$ it is basically L2 loss thus lemma from 4.1 is still valid with slight modification.

Regularization was used only during the robustness experiments and was selected to be proportional to the noise level, $\lambda = 0.01 \times \text{STD}(\xi)$. Since XiEff is optimized for each NFI from the dataset and all NFI data is within a batch, we chose to use the second-order L-BFGS algorithm with strong Wolfe conditions Wolfe (1969). Optimization was limited to 20 steps for regular experiments; for robustness measurement experiments we stopped optimization at 10 steps. The entire optimization process for a single NFI (grid 100×100) consumed approximately 1–2 min on 12GB GPU. It is significantly faster than Chen et al. (2020); Chen & Dal Negro (2022) and comparable to Saba et al. (2022). As the quality metric we chose the Intersection over Union (IoU) between the particle shape and the shape of the thresholded norm of XiEff. The IoU is a commonly used metric in computer vision and image segmentation tasks, providing a measure of the overlap between the predicted and ground truth segmentations thus well-suited for the problem under consideration. The threshold for every NFI was chosen to maximize the IoU. In our case, it quantifies the similarity between the optimized XiEff representation and the actual particle shape, enabling us to evaluate the interpretability of the learned representation. The table 1 presents the IoU metrics for both dataset variants with constant and random external fields for each category of shapes: ellipses, rectangles, convex, unions, characters, and the average across the entire dataset.

The IoU values (table 1) showcase the method's proficiency with simple geometric shapes like ellipses and rectangles, achieving excellent performance across both datasets. For more complex

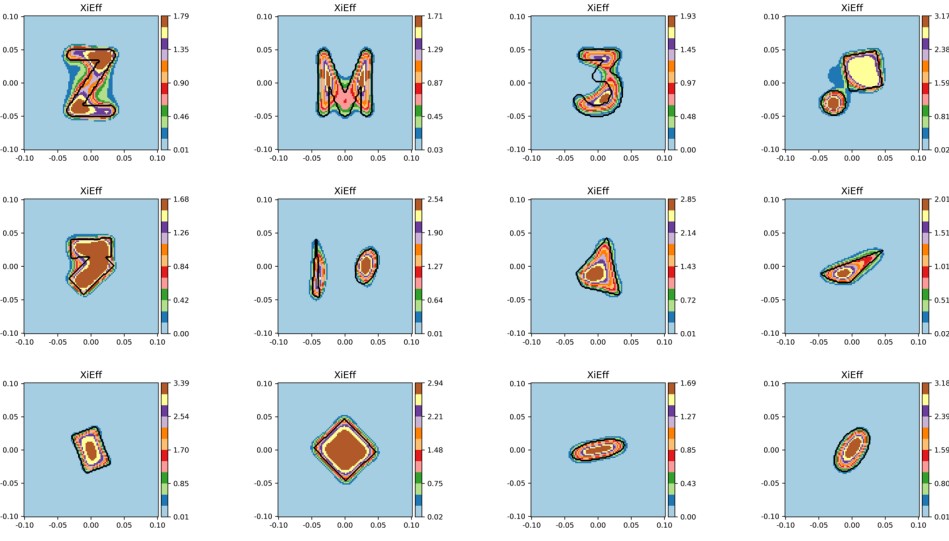

Figure 5: Random dataset: Xieff.

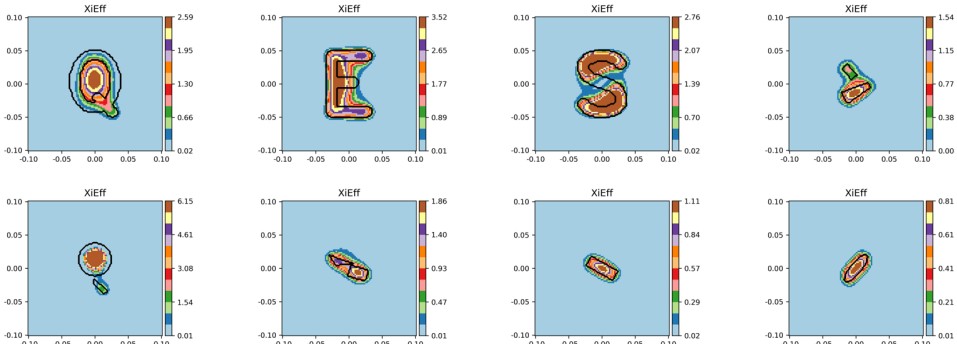

Figure 6: Random dataset: Hard cases.

categories like convex shapes and shape unions, the method still exhibits promising results by faith-
fully modeling the underlying structure. Overall, the IoU metrics highlight exceptional performance
with simple shapes and remarkable generalization to increasing complexities.

Additionally, we conducted experiments regarding robustness to noise. These experiments involved
adding Gaussian noise to the NFIs and measuring the IoU for the random variation of the dataset.
The results of this modeling are presented in Figure 3, which displays the IoU versus the level
of the noise injected. The level of the standard deviation of the NFI within the dataset variation
without noise is also displayed. As we can see, the proposed method is robust because the IoU
metric degrades slowly, even in cases where the noise level is significantly higher than the standard
deviation of the dataset NFIs. This robustness is a crucial property for practical applications, where
measurement noise is inevitable, and it demonstrates the effectiveness of the proposed approach in
handling noisy observations.

Figure 4 displays the visualization of the constant dataset variant NFI with the corresponding XiEff
representation. As you can see, the NFI images of different particles are very similar to one another
and are not explainable. However, the corresponding XiEff representation (the result of optimization
from the NFI data) is highly similar to the particle shape, providing excellent interpretability for the
NFI images. This interpretability is a key advantage of the proposed method, as it enables a better
understanding of the underlying physical phenomena and the characteristics of the particles under
investigation.

The random dataset variant is displayed in Figure 5 with the XiEff representation but without the NFIs due to the low visual clarity of the NFIs with randomly changed external fields at every probe position. As you can see, it works well even for more complex particle shapes, demonstrating the versatility and generalization capabilities of the proposed approach.

The method fails in two types of cases (see figure 6): first, thin or small-sized particles, and second, particles with holes. We can assume that the first type of failure case may potentially be partially improved by changing the neural network architecture, optimization process, or even by increasing the level of discretization. However, the second type of failure case cannot be easily resolved due to the physical limitations, as a similar problem exists in classical optics as well. These failure cases highlight the need for further research and improvements, particularly in addressing the limitations posed by complex particle geometries.

## 7 CONCLUSION

In this work, we introduced a novel computer vision algorithm—the XiEff representation—that leverages neural fields to model the effective susceptibility tensor in near-field optics. By integrating this representation into the Lippmann-Schwinger integral equation, we developed an unsupervised method to reconstruct the effective susceptibility directly from imaging data, operating without the need for labeled or external data. Also the proposed algorithm is without any method specific hyperparameters and as result efficient in tuning.

Our experimental validation on synthetic data demonstrated high accuracy, with the method performing well on both simple and complex shapes and achieving high Intersection over Union (IoU) scores. Theoretical analysis and empirical evaluations confirmed the method's robustness to measurement noise, ensuring reliable convergence even with noisy observations. This robustness is crucial for practical applications where measurement noise is inevitable.

Furthermore, we proposed a new scanning strategy for SNOM, where the external field is randomized at each probe position. Our experiments with this random dataset showed an increase in the quality of the inverse problem solution. This approach appears to be universal and could potentially improve the performance of other inverse algorithms in similar domains.

The optimized XiEff representation offers improved interpretability by providing a physically meaningful model directly connected to the optical characteristics of objects. It effectively addresses the limited visual clarity of raw near-field images, which are often challenging for human perception to interpret.

In addition, we have developed a comprehensive synthetic dataset that will be publicly released, potentially serving as benchmark for inverse problems in near-field optics and PINNs.

Our successful application of an integral equation approach within the PINNs framework is particularly noteworthy, as such cases are relatively rare. This achievement opens new avenues for incorporating integral equations into physics-informed machine learning, potentially benefiting a wide range of physical problems where integral formulations offer advantages over differential equations.

Future research opportunities include exploring more advanced neural network architectures and optimization techniques from the PINNs perspective to further enhance the performance of the XiEff representation. From the near-field optics standpoint, extending the method to three-dimensional geometries, objects on or under surfaces, anisotropic and nonlinear materials, and applying the method to real SNOM data are promising directions. Given that the Lippmann-Schwinger equation is generally valid, the proposed algorithm could be applied to other wave regimes, such as radio frequencies, and could be expanded from the near-field to the far-field zone.

In conclusion, our theoretical research, focused on practical outcomes, makes a significant advancement in expanding computer vision's domain into near-field optics modality. The XiEff representation pushes the boundaries of Physics-Informed Computer Vision, demonstrating how interdisciplinary approaches can transform scientific imaging, converting complex physical measurements into comprehensible, machine-learned representations which are interpretable from a physical perspective.

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
