# OpenReview forum: "XiEff Representation for Near-Field Optics"
_ICLR.cc/2025/Conference — ICLR 2025 Conference Withdrawn Submission_

### Official Review · Reviewer_f4u8 · 2024-10-31

**Soundness:** 2
**Presentation:** 3
**Contribution:** 2
**Rating:** 5
**Confidence:** 3

**Summary:**

This article addresses the issue of lacking visual clarity and interpretability in images captured by scanning near-field optical microscopy. Based on the principle of neural radiance fields, the authors propose an effective susceptibility tensor parametrization method grounded in neural fields. This method enhances the interpretability of imaging and the morphological accuracy of micro-objects.

**Strengths:**

1. The article introduces a new representation method called XiEff. This approach is inspired by Neural Radiance Fields (NeRFs) in computer vision and Physics-Informed Neural Networks (PINNs).

2. The authors have integrated the XiEff representation into the Lippmann-Schwinger (LS) integral equation framework used in near-field optics, developing an optimization strategy that can directly reconstruct the effective susceptibility distribution from Near-Field Imaging (NFI) data.

3. An unsupervised optimization framework is proposed that can directly reconstruct the effective susceptibility distribution from NFI data. The authors claim that this method improves the practicality and efficiency of near-field optical analysis.

**Weaknesses:**

Weakness：
I have multiple concerns and questions about this paper, please see the detailed comments and suggestions.

**Questions:**

1. The authors did not provide a systematic introduction to the relevant research background of this study, i.e., the most pertinent previous works were not discussed.
2. Most of the references cited in the paper are relatively old; the authors need to incorporate comparisons with more recent research content.
3. The "interpretability" emphasized in this task requires a clear definition from the authors.
4. Ground Truth should be separately listed in Figure 4.
5. The experimental section is rather thin; there are no ablation studies or evaluations of model parameters.
6. All the data used in the paper are synthetic samples; real-world microscopic data were not utilized.
7. Whether the paper fits within the scope of ICLR is questionable, as it does not delve deeply into concepts from machine learning or artificial intelligence.

---

> ### Author Response · Authors · 2024-11-22
>
> We thank the reviewer for their thoughtful feedback and valuable suggestions, which have significantly improved our paper. We have addressed your concerns as follows:
>
> We have fully rewritten the **Introduction** and **Conclusion** sections and added a new **Related Work** section to provide a systematic introduction to the research background of our study. This includes discussions of the most pertinent previous works, incorporating comparisons with recent research to situate our work within the current state of the field. We have also extended our citations to include more recent studies, ensuring that our references reflect the latest advancements.
>
>
> **•	Interpretability:** We have provided a clear definition in the revised manuscript. The interpretability of our method lies in the XiEff representation being well-defined with rigorous formulas, where each component of the tensor XiEff has a clear physical meaning. This means that our algorithm is interpretable and does not require labeled data. In contrast, traditional brute-force computer vision algorithms like vanilla UNET or diffusion models require large amounts of training data and function as black boxes, lacking this level of interpretability.
>
> **•	Experimental Section and Ablation Studies:**  Our main commitment is to expand computer vision into the new modality of near-field optics, focusing on applied results that could be useful in this field.
>  We have included experiments on shape reconstruction, robustness analysis, and the SNOM scanning strategy adopted for the inverse algorithm. Our research presents a computationally efficient, robust, hyperparameter-free method suitable for near-field optics and other domains described by Maxwell's equations. Research on optimal architectures of XiEff neural fields will be part of future work.
>
> We acknowledge that ablation studies are typically required when a machine learning work is based on modifying or blending several approaches or ideas. In our case, we have employed a successful MLP architecture from NeRFs without modification. Tuning it did not reveal novel insights, so we decided not to include this topic. Our proposed method is atomic without subcomponents and is hyperparameter-free because it does not introduce new hyperparameters related to the method itself.
>
> However, we have provided three studies which could be considered as an ablations regarding to our proposed algorithm:
>
> 1) 	Noise Conditions: We experimentally measured the influence of noise on the algorithm's accuracy and introduced regularization, which was used only during robustness experiments.
>
> 2) 	Influence of SNOM Scanning Strategy: We studied the impact of the SNOM scanning strategy on the inverse algorithm. We have proven that random changes in the pumping field of the microscope at every probe position increase the quality of the proposed method. This result could be general for many inverse algorithms.
>
> 3) 	Performance Across Shape Complexities: We measured the method's performance for cases with increased complexity, including samples of simple forms (Ellipses, Rectangles, Convex shapes), Unions (Pairs of simple forms), and Characters. This research allowed us to identify challenging cases and limitations of the method.
>
> **•	Synthetic Data Usage:** We acknowledge that all the data used in the paper are synthetic samples. We chose to use synthetic data due to the practical challenges and costs associated with obtaining real-world microscopic measurements of nanosized particles with diverse shapes. The ablation studies performed with this synthetic dataset would be impossible to conduct with comparable quality using real-world data due to the practical limitations and challenges involved. Testing on real-world microscopic data is considered valuable future work.
>
> **•	Scope of ICLR:** We believe that our work fits within the scope of ICLR as it contributes to machine learning and artificial intelligence by integrating neural fields and physics-informed approaches to solve inverse problems in near-field optics. By expanding computer vision into this new modality, we demonstrate how interdisciplinary methods can advance scientific imaging. Our work pushes the boundaries of Physics-Informed Computer Vision (PICV) and boundary "Learning Representation" in general. Our work matches several topics listed for ICLR 2025.
>
> We hope that these revisions and clarifications address your concerns and enhance the quality of our paper.

---

> > ### Comment · Reviewer_f4u8 · 2024-11-26
> >
> > For the data issue, I still have doubts. There is generally a domain gap between synthetic data and real data. It is essential for the authors to validate their work using real data in subsequent studies.

---

> ### Author Response · Authors · 2024-11-28
> **Answer regarding domain gap between synthetic data and real data.**
>
> Thank you for your question.
>
> We have demonstrated the robustness of the XiEff representation to noise both theoretically and through extensive experiments. This ensures that the algorithm remains reliable even under conditions similar to real experiments.
>
> The XiEff representation belongs to a class of algorithms that do not rely on labeled or external data. For such algorithms, the domain gap between synthetic and real-world data is a less significant issue compared to methods trained on specific datasets and then applied to new test data. Unlike traditional machine learning models, these algorithms are not prone to overfitting to a specific training set or domain.
>
> Similar algorithms, such as t-SNE and NeRFs, also do not rely on external training data and have been shown to work effectively across diverse domains, including both synthetic and real-world data. The XiEff representation can be considered a physically rigorous generalization of NeRFs. While NeRFs are successful in domains where geometric optics is valid, we expect the XiEff representation to perform well in domains governed by the selected Green's function and Lippmann-Schwinger equation.
>
> Historically, methods like NeRFs were initially developed and tested primarily on synthetic datasets, even though real-world images were readily available ( https://arxiv.org/pdf/2003.08934 ). Similarly, t-SNE was first validated on relatively simple datasets like MNIST ( https://jmlr.org/papers/volume9/vandermaaten08a/vandermaaten08a.pdf ), with many follow-up works relying only on synthetic and simple data. In the case of near-field optics, where NFI images are challenging and expensive to obtain, the usage of synthetic data is a practical and reasonable compromise, particularly for a theoretical conference like ICLR.
>
> In summary, we propose a visualization tool for the near-field optics domain, with potential broad applicability to other domains using Maxwell's equations. Validation on real SNOM data will be a significant focus of future work.

---

### Official Review · Reviewer_1BZx · 2024-11-02

**Soundness:** 1
**Presentation:** 2
**Contribution:** 1
**Rating:** 3
**Confidence:** 4

**Summary:**

**Summary**
This submission proposes the use of Physics-Informed Neural Networks (PINNs) for processing data from Scanning Near-Field Optical Microscopy (SNOM). While the motivation is clear, the contribution is poor, with significant literature missing regarding the application of PINNs for inverse problems in nano-optics, as well as specific applications to SNOM data.

**Clarity**
- The work would benefit from a dedicated Related Work section, which should discuss prior research on PINNs and their application in nano-optics optimization, as well as relevant papers on near-field optics.

**Significance and Originality**
(-) The originality of this paper is limited, especially as it omits relevant literature on its proposed contribution. Notably, the foundational paper by Chen et al. (2020), “Physics-informed neural networks for inverse problems in nano-optics and metamaterials,” is not discussed. Additionally, the paper specifically addressing this problem by Chen et al. (2022), “Physics-informed neural networks for imaging and parameter retrieval of photonic nanostructures from near-field data,” is missing from the discussion. The 2020 paper has 500 citations, while the 2022 paper has 40 citations, making these omissions a critical oversight.

**Results**
(-) By 2024, it is insufficient to present PINN-related work at a top conference using only 2D toy results, especially considering the breadth of related research. The authors should either address the failure modes of previous methods or propose a novel methodology with robust validation.

**References:**
- Chen, Yuyao, et al. "Physics-informed neural networks for inverse problems in nano-optics and metamaterials." *Optics Express*, 28.8 (2020): 11618-11633.
- Chen, Yuyao, and Luca Dal Negro. "Physics-informed neural networks for imaging and parameter retrieval of photonic nanostructures from near-field data." *APL Photonics*, 7.1 (2022).

**Strengths:**

Introducing "neural fields + optics" into ICLR is an interesting endeavor.

**Weaknesses:**

stated above.

**Questions:**

Stated above.

---

> ### Author Response · Authors · 2024-11-22
>
> We thank the reviewer for their thoughtful feedback and have made significant revisions to address the concerns. We have fully rewritten the Introduction and Conclusion, added a dedicated "Related Work" section, and provided more clarification regarding the realism of our dataset. Additionally, we have added the papers by Chen et al. (2020, 2022) to the Related Work section. These papers are also considered in the broader context of Physics-Informed Computer Vision, as cited in Banerjee et al. (2024).
>
> **•	Regarding originality and contribution:**
>
> Our approach differs fundamentally from that of Chen et al. While they used traditional PINNs embedding Maxwell's differential equations (system of 4 equations) and boundary conditions into the loss function—leading to complex training procedures and hyperparameter tuning—our method is based on the integral Lippmann-Schwinger equation. This integral formulation implicitly incorporates physics through the Green's function and handles boundary conditions inherently, eliminating the need for explicit boundary condition terms and excess hyperparameters. As a result, our algorithm is hyperparameter-free and significantly more computationally efficient.
> For instance, Chen et al. (2022) reported that their 3D reconstruction of a spherical particle on a computational grid of 50×50×30 (75000 points) took about 10 hours on a 1080Ti GPU. In contrast, our XiEff representation algorithm for a 2D case with a 100×100 (10000 points) grid takes approximately 1–2 minutes on a standard GPU (e.g., Google Colab GPU with 12 GB). This demonstrates the computational efficiency of our approach.
>
> **•	Regarding experimental validation**
>
> Chen et al.'s methods were validated on simpler shapes and tasks, such as scalar property retrieval and spherical object reconstruction, without extensive benchmarking according to modern machine learning standards. To address this, we have developed and will release a comprehensive synthetic dataset covering a wide variety of complex shapes, enabling thorough validation and reproducibility. This approach follows best practices from mature machine learning fields like computer vision and NLP.
>
> We acknowledge that our experiments were conducted on synthetic data. Creating a comprehensive real SNOM dataset with diverse shapes is a significant technological challenge currently beyond our capabilities. However, our synthetic dataset allows us to validate our method on shapes that are difficult to reproduce experimentally. Since SNOM data is typically 2D, we believe the domain shift is manageable. Complexity scaling to the 3D case primarily involves computational considerations rather than fundamental methodological changes. The similar challenges has  already resolved in  more mature areas like Nerfs which now successfully scaled to 4D and works with images and even video.
>
>
>
> Our work has yielded practical advancements, including an unsupervised reconstruction method robust to noise and a novel scanning strategy for near-field imaging specialized for inverse problems.
> In conclusion, our work offers a more efficient and practical approach by leveraging integral equations within the PINNs framework, differing significantly from prior methods in both methodology and validation. We believe these contributions address the reviewer's concerns and demonstrate the originality and practical potential of our approach.

---

> > ### Comment · Reviewer_1BZx · 2024-11-25
> > **I raise the score to 3 but I still think the novelty is poor and the draft needs more work**
> >
> > Thanks for your reply. As I see from the revised version. The experiments are still not enough. As I said before, having only 2D sythnetic data is not enough. As well, given the work from Chen et al. It is hard to accept an incremental work on this topic.
> >
> > Moreover, I do suggest you improve your writing, and particularly, incorporate comparison to Chen et al in the paper, but not just answered in the openreview.
> >
> > The related work section is still very limited. You should have a bit thorough analysis on the papers in the related topics, such as including paper "Physics-informed neural networks for diffraction tomography".

---

> > > ### Author Response · Authors · 2024-11-28
> > > **Improved manuscript and added supplementary materials**
> > >
> > > Thank you for raising the score and for your constructive feedback. We greatly appreciate your thoughtful comments and suggestions.
> > >
> > > **Experiments:**
> > >
> > > Our implemented algorithm indeed allows for additional experiments. However, the 10-page limitation of the manuscript constrained the inclusion of more details. As a compromise, we have uploaded supplementary materials that include a visual log of the XiEff representation algorithm for the random dataset case. These materials contain the final optimized XiEff images with the corresponding electromagnetic field in the root folder, as well as visualizations of the optimization process for each dataset sample, with XiEff images displayed after every optimization step. These supplementary materials demonstrate the method’s good convergence, often achieving reliable results within 5–15 optimization steps, proving the efficiency of the algorithm.
> > >
> > > Additionally, a comparable case is seen in the initial NeRF paper (https://arxiv.org/pdf/2003.08934), which, despite having existing benchmarks and a 15-page allowance, presented a similar number of experiments as our work. This further supports the experimental design and scope of our paper.
> > >
> > > **Domain Shift Between Synthetic and Real SNOM Data:**
> > >
> > > Please see our detailed response to Reviewer f4u8. Additionally, our method has demonstrated robustness to domain shifts in the shape of the object similiar to NeRFs.
> > >
> > > **Comparison with Chen et al.'s Work:**
> > >
> > > We have revised the manuscript once more to include additional comparisons with Chen et al.’s studies. Details about performance comparisons and differences in hyperparameters can now be found in the "Related Work" and "Experiments" sections.
> > >
> > > Regarding datasets, as elaborated in the "Related Work" and "Dataset" sections, Chen et al.'s studies were not focused on shape retrieval. Their dataset design allowed for unintended leakage of the particle's form due to measurements of field taken directly near the surface of object without proper padding. In our dataset, we used a mask to cover the particle, ensuring that the neural network received no direct information about the particle's shape.
> > >
> > > Another key algorithmic distinction, which was not fully emphasized in the paper due to manuscript freeze, involves what is reconstructed. Chen et al. reconstructed both the field ${E_k}$ and the susceptibility $\chi_{jk}$ distribution within the object. This dual reconstruction inherently adds complexity. Our method, on the other hand, reconstructs the effective susceptibility:
> > > $$
> > > X_{jk} = \chi_{jk}\frac{E^{(0)}_k}{E_k}.
> > > $$
> > > By doing so, we provide semi-analytical surrogate for the forward solver which is  a physically rigorous, hyperparameter-free, differentiable, and computationally efficient:
> > > $$
> > > E(\vec{r}) = E^{(0)}(\vec{r}) -  \int G(\vec{r}, \vec{r'}) X(\vec{r'}) E^{(0)}(\vec{r'}) d\vec{r'}
> > > $$
> > > This significantly reduces computational cost, with XiEff completing reconstructions in about 2 minutes , compared to the 10 hours reported by Chen et al. Our algorithm's efficiency also allows the use of computationally intensive optimizers like LBFGS.
> > >
> > > Thus, we argue that our work is a significant advancement in "representation learning" rather than an incremental improvement.
> > >
> > > **Inclusion of Suggested References:**
> > >
> > > We have included the paper "Physics-informed neural networks for diffraction tomography" in the Related Work section. In this study, the authors used a UNet-based physics-informed neural network as a forward solver which used in iverse reconstruction as subcomponent.  This UNet sover was firstly pre-trained on a synthetic dataset of biological cell objects and then fine-tuned on real data.  As a result, its applicability is limited to shapes represented in the training dataset. By contrast, our method relies on a physically rigorous solver, making it robust to domain shifts in shape or noise, similar to NeRFs. Also, our approach is trained in an end-to-end fashion.
> > >
> > > Thank you once again for the references and suggestions. In summary, our work demonstrates significant potential for developing unsupervised algorithms similiar NeRFs and t-SNE tailored for domains described by Maxwell's equations.

---

### Official Review · Reviewer_XNDG · 2024-11-03

**Soundness:** 2
**Presentation:** 2
**Contribution:** 1
**Rating:** 3
**Confidence:** 4

**Summary:**

The paper presents a method, termed XiEff Representation, for reconstructing particle shapes from near-field optical images obtained through Scanning Near-Field Optical Microscopy. The approach leverages neural fields inspired by Neural Radiance Fields (NeRF) and Physics-Informed Neural Networks (PINNs) to parameterize the effective susceptibility tensor within the Lippmann-Schwinger equation framework. The authors propose this technique as an interpretable model for particle shape reconstruction, particularly under noisy conditions, using an unsupervised optimization framework.

**Strengths:**

This approach adapts NeRF and PINNs, for effective shape reconstruction in nano-optics; and shows robustness in noise on a synthetic dataset with various particle geometries.

**Weaknesses:**

While the XiEff representation is positioned as an improvement over traditional methods, the paper lacks a rigorous quantitative comparison with existing approaches for solving inverse problems in nano-optics, such as traditional discretization, iterative, or diagram-based solutions. A discussion on seminal works such as Chen et al. (2020, 2022), which apply PINNs to inverse problems in nano-optics, is also absent.

**Questions:**

1. While the proposed XiEff method shows good shape reconstruction, it is not clear how much is improved compared to traditional reconstruction methods. Better to include these in the related work section, as well as add a reconstruction result comparison.
2. The paper only presents results on synthetic data. How does this technique work on real SNOM data?
3. Similarly, should also acknowledge and discuss prior works on PINNs in nano-optics/near-field optics to better position itself within the existing literature.

---

> ### Author Response · Authors · 2024-11-21
>
> Thank you for your insightful feedback on our manuscript. We have made significant revisions, including a complete rewrite of the Introduction and Conclusion, the addition of a Related Work section, and improved a Dataset section.
>
> Regarding your first point, we have incorporated discussions of Chen et al. (2020, 2022) in the Related Work section to better position our method within the existing literature. We also reference Banerjee et al. (2024) where these works mentioned in the broader context of Physics-Informed Computer Vision. These prior works applied PINNs to inverse problems in nano-optics but focused on simpler shapes and tasks, such as reconstructing scalar properties like electric permittivity and magnetic permeability, or the shape of spherical objects. In contrast, our method addresses more complex geometries and provides an unsupervised solution that is robust to noise. To enhance validation and reproducibility, we are releasing our synthetic dataset publicly, allowing for benchmarking of similar solutions in near-field optics inverse problems using PINNs.
>
> Regarding your second point, we acknowledge that our experiments were conducted on synthetic data. Creating a comprehensive real SNOM dataset with diverse shapes is a significant technological challenge and currently beyond our capabilities. However, the synthetic dataset enables us to validate our method on a wide variety of shapes that are difficult to reproduce experimentally. Since SNOM data is typically 2D, we believe the domain shift between our synthetic data and real SNOM data is manageable. Despite this limitation, our work has yielded practical advancements. We developed an unsupervised reconstruction method that remains robust even with noisy data, and we proposed a novel algorithm for scanning near-field imaging specialized for inverse problems. These contributions demonstrate the practical potential of our approach and address some of the challenges in the field.
>
> We hope these revisions address your concerns and clarify the significance of our contributions.

---

> > ### Comment · Reviewer_XNDG · 2024-11-30
> > **Improved manuscript but lack of contributions remain the same**
> >
> > Thanks for the response and for submitting the revised version. I appreciate the current related work section; but the efficiency of the method is only validated with an IoU score, without any quantitative comparison with other reconstruction methods mentioned in the related work. At the same time, I don't think the authors have convinced me of not applying this method to a real SNOM dataset, which would improve their contributions in this work.

---

> > > ### Author Response · Authors · 2024-12-01
> > > **use of IoU, comparison with existing methods, real SNOM data**
> > >
> > > Thank you for your valuable feedback. We appreciate your time and constructive comments.
> > >
> > > **Regarding the use of IoU and additional metrics:**
> > >
> > > We have added supplementary materials that include logs with additional metrics such as the loss function and L2 loss outside the object region. These provide deeper insights into the optimization process and convergence behavior.
> > >
> > > While regression metrics like L1, L2, or MSE are common in evaluating PINNs (as seen in Chen et al., 2020, 2022; Saba et al., 2022), inverse problems in near-field optics are inherently ill-posed. For shapes with holes or internal structures (see letter Q at Figure 6: Random dataset: Hard cases), it's challenging to reconstruct internal structures within materials because we cannot obtain information from inside the object using optical methods. Thus, regression metrics may not fully capture reconstruction effectiveness when only surface reconstruction is achievable.
> > >
> > > Focusing on surface reconstruction, we considered metrics from semantic segmentation in computer vision, such as pixel accuracy, Dice coefficient, and Intersection over Union (IoU). We chose IoU because it effectively measures the overlap between the predicted and actual particle shapes, providing a meaningful assessment of reconstruction accuracy.
> > >
> > > By establishing a benchmark dataset with the IoU metric, we aim to provide a standardized evaluation framework for the optical community. This will facilitate fair comparisons and promote the development of improved methods in Physics-Informed Computer Vision (PICV) and PINNs inverse problems.
> > >
> > > **Regarding quantitative comparison with existing methods:**
> > >
> > > Our method fundamentally differs from existing works like those by Chen et al. Their approaches rely on dual optimization of both the electric field and susceptibility distribution, adding complexity and computational overhead. In contrast, we directly optimize the effective susceptibility (XiEff), offering a hyperparameter-free (Chen et al., 2022, contains 4 hyperparameters), differentiable, and computationally efficient surrogate for the forward solver.
> > >
> > > Performance-wise, our method significantly reduces computational time and resources. Chen et al. (2022) reported that their 3D reconstruction took about 10 hours on a 1080Ti GPU for a grid of 75,000 points. Our method achieves reconstruction on a 2D grid of 10,000 points in approximately 1–2 minutes on a standard GPU. This efficiency allows us to use optimizers like L-BFGS for faster convergence.
> > >
> > > Moreover, Chen's experiments were not primarily focused on reconstructing complex shapes and used measurements taken directly at the object's surface, leaking shape information into data. Our method reconstructs shapes from measurements taken at a distance (in shape of rectangular mask), making the inverse problem more challenging and the reconstruction more meaningful.
> > >
> > > Chen's papers have not released the code to GitHub, requiring full reimplementation from scratch. Their method has many hyperparameters to tune along with significant computational power requirements, making it hard to reproduce within our team.
> > >
> > > By introducing a benchmark dataset with evaluation metrics like IoU, we enable researchers to compare methods directly without independently reproducing other solutions, facilitating progress through fair and consistent evaluations. We will release our solution to GitHub to ensure broader access and reproducibility.
> > >
> > > **Regarding application to real SNOM datasets:**
> > >
> > > We acknowledge the importance of validating our method on real data. However, using synthetic data allows us to thoroughly test and validate our method on a wide range of complex shapes that are technologically challenging to fabricate and measure at the nanoscale, demonstrate robustness to noise and introduce proper metrics IoU instead conventional PINNs regression metrics.
> > >
> > > Furthermore, the theoretical framework used in our work has previously been successfully applied to real SNOM data, demonstrating its practical applicability. See same examples: Lozovski (2010) "The effective susceptibility concept in the electrodynamics of nano-systems", Martin et al. (1994) "Iterative scheme for computing exactly the total field propagating in dielectric structures of arbitrary shape", and Paulus & Martin (2001) "Light propagation and scattering in stratified media: a Green’s tensor approach". These studies assure us that our framework is sound and applicable to real SNOM datasets.
> > >
> > > Thank you again for your feedback. We hope our clarifications address your concerns and highlight the contributions and potential impact of our work in near-field optics and inverse problems.

---

### Official Review · Reviewer_qMTD · 2024-11-03

**Soundness:** 1
**Presentation:** 3
**Contribution:** 1
**Rating:** 3
**Confidence:** 4

**Summary:**

This paper tackles the problem of effective susceptibility distribution reconstruction from near-field imaging data. The idea is to use the physics-informed neural networks (PINN) to solve the Lippman-Schwinger equation given the external field observation from the SNOM probe.

**Strengths:**

In general, the paper is clearly written, though there are some formatting issues (such as the citation format and the section fragments) should be fixed.

**Weaknesses:**

The paper, however, can be viewed as an exercise of PINN for the specific near-field imaging tasks without any algorithmic innovations. Note, ICLR typically emphasizes novelty on the algorithmic side. Simply adopting an existing (and well-known) approach to a highly domain-specific problem (like the near-field imaging) would not be recommended for ICLR publication.
I would suggest the authors further refine their paper, focus on the experiments (the experiments conducted in this paper are too toy to be attractive), and submit it to an optics/photonics journal for next-round evaluation.
Particularly, a 3-D experimental setup with non-diagonal effective susceptibility would be interesting to be explored, in contrast to the 2D, diagnonal cases tested in the paper.

**Questions:**

See above

---

> ### Author Response · Authors · 2024-11-21
>
> Thank you for your thoughtful feedback. We have made significant revisions to our manuscript, including a complete rewrite of the Introduction and Conclusion sections, added a detailed Related Work section, and provided additional clarification regarding the realism of our dataset.
>
>
> **1. Regarding ICLR and Innovations:**
>
> Our work aligns with several key topics of interest at International Conference on Learning Representations (ICLR) https://iclr.cc/Conferences/2024/CallForPapers :
>
> •	Unsupervised Representation Learning: We developed the XiEff representation, a novel unsupervised Physics-Informed Computer Vision algorithm that does not require labeled data.
>
> •	Representation Learning for Computer Vision: Our method extends computer vision into the domain of near-field optics at the nanoscale, a new modality with significant potential impact on modern technology.
>
> •	Datasets and Benchmarks: We created a synthetic dataset that can serve as a public benchmark for near-field optics and inverse problems in PINNs.
>
> •	Neurosymbolic & Hybrid AI Systems: Our approach integrates physical principles, specifically the Lippmann-Schwinger integral equation, into neural network optimization. Method is hyperparameter free and computationally efficent in oposite to approached based on PDEs.
>
> •	Applications to Physical Sciences: The developed equations and algorithms are generic and applicable to other fields described by Maxwell's equations, including radiowaves [Zhang et al. (2024)] in near-field and far-field zones, and are valid even for geometric optics.
>
> We believe our contributions offer valuable insights and tools that can benefit a wide range of machine learning areas within the ICLR community.
>
>
> **2. Regarding the Dataset and Experiments:**
>
> We focused on 2D geometric cases because generating real-world NFI data involves creating nanosized particles and measuring them, which is both time-consuming and costly—especially when a diverse range of shapes is required. A synthetic 2D dataset is more practical to start with, as it is easier to visualize, provides diversity, and aligns with the typical 2D NFI images measured by SNOM. This approach is consistent with the development history of computer vision, where problems are first tackled in 2D before extending to 3D.
>
> Our choice of isotropic susceptibility is well-justified, as most common dielectric materials exhibit only weak anisotropic responses at optical frequencies [Landau et al. (2013); Boyd et al. (2008)]. For our 2D geometry, system symmetry further supports this treatment, maintaining physical relevance while keeping the inverse problem tractable.
>
>
>
> Despite using a synthetic dataset, our primarily theoretical work has yielded practical results:
>
> •	Unsupervised Reconstruction: The XiEff representation allows for the reconstruction of optical properties of objects without labeled data.
>
> •	Robust Optimization: The optimization process is stable even in the presence of noise.
>
> •	New SNOM Scanning Algorithm: We proposed a novel algorithm for scanning NFI images tailored for inverse problems.
>
> •	Benchmark Introduction: We introduced a new benchmark that can be utilized in both near-field optics and inverse tasks in PINNs.
>
>
> We appreciate your consideration and are confident that our revisions have strengthened the manuscript.

---

### Author Response · Authors · 2024-12-02
**Validation question. Contributions across research domains.**

We would like to address common questions raised regarding our work:

**1) Validation on Synthetic vs. Real SNOM Data**

Obtaining real SNOM data with diverse and complex nanoscale shapes is challenging due to technological and practical constraints, and such data is not readily accessible. As a result, we conducted our research using a synthetic dataset. This approach allowed us to:

- Theoretically investigate and experimentally validate the robustness of our method under controlled conditions.

- Demonstrate that our method effectively reconstructs complex shapes that are difficult to fabricate nanotechnologically.

- Identify and analyze hard cases, introducing appropriate metrics (IoU) for validation.

- Develop and test a novel NFI scanning algorithm tailored for inverse problems.

Our proposed method does not rely on external data or pre-training, making it inherently robust to domain shifts, such as those between synthetic and real data. Algorithms in the same class, such as t-SNE and NeRFs, have demonstrated robustness and universality across different domains.

Moreover, our work is grounded in theoretical physics frameworks that have been extensively validated with real near-field optics data in many researches conducted in prior decades. This foundation gives us confidence that our method would perform well on real SNOM data, and we consider testing on real data an important direction for future work.

**2) Contributions Across Different Research Domains**

Given the multidisciplinary nature of our work, we would like to emphasize our contributions in the context of various research areas:

*PDE Mathematical Theory and PINNs:*

While much of the PINNs community focuses on partial differential equations (PDEs), our work leverages integral equations, specifically the Lippmann-Schwinger equation. Together with NeRFs(probably, the most successful PINN models), our approach showcases the effectiveness of models based on integral equations, expanding the mainstream focus beyond PDEs.

*Theoretical Physics and Applied PINNs:*

By integrating the concept of effective susceptibility into the Lippmann-Schwinger equation, we introduced a surrogate for the forward modeling solver. This surrogate is:

- Robust and physically rigorous, making it hyperparameter-free.

- Implicitly handles boundary conditions, simplifying the modeling process.

- Computationally efficient, allowing for faster and more scalable computations.

Our approach enables efficient modeling of inverse problems by optimizing the effective susceptibility  directly, rather than jointly optimizing both the field and susceptibility. This methodology is universal and can be applied to systems described by Maxwell's equations, including near-field and far-field regimes, optical and radio waves, and even geometric optics.

*Representation Learning:*

We introduced a novel unsupervised algorithm—the XiEff representation. Our algorithm has several advantages over existing methods:

- Avoids dual optimization, simplifying the computational process.

- Hyperparameter-free, reducing the need for extensive tuning.

- Robust to noise and does not rely on labeled or unlabeled external data, enhancing robustness to domain shifts.

- Provides interpretable results

- Computationally efficient (2min vs 10h), even allowing us to employ L-BFGS for optimization.

In essence, we propose a tool similar to t-SNE but adapted for systems governed by Maxwell's equations.

*Computer Vision (Physics-Informed Computer Vision):*

We extended computer vision into the new modality of near-field optics, a significant step in Physics-Informed Computer Vision. We established a new dataset and introduced appropriate validation metrics (such as IoU) tailored to this domain, moving beyond traditional PINNs regression metrics. We also plan to release our baseline solution code on GitHub, facilitating further research and development. The proposed XiEff representation algorithm is a physically rigorous generalization of NeRFs, derived from Maxwell's equations.

*Near-Field Optics, SNOM, Photonics:*

We bring the latest advancements in representation learning to the existing theoretical framework for modeling near-field optics. This framework, based on the Lippmann-Schwinger equation and Green's function formalism, has been extensively tested with real SNOM data over the past decades. Our work enhances near-field imaging by improving interpretability and efficiency. Additionally, we introduced an algorithm for specialized near-field imaging tailored for inverse problems.

We believe that our work makes significant contributions across these domains, offering novel insights and tools that bridge theoretical physics, machine learning, and practical applications in near-field optics.

Thank you again for your engagement and consideration.

---

### Note · Authors · 2025-01-22

I have read and agree with the venue's withdrawal policy on behalf of myself and my co-authors.